# An open-sourced, web-based application to analyze weekly excess mortality based on the Short-term Mortality Fluctuations data series

**László Németh**[1]*, **Dmitri A. Jdanov**[1,2], **Vladimir M. Shkolnikov**[1,2]

**1** Max Planck Institute for Demographic Research, Rostock, Germany, **2** National Research University Higher School of Economics, Moscow, Russia

* nemeth@demogr.mpg.de

## Abstract

The COVID-19 pandemic stimulated the interest of scientists, decision makers and the general public in short-term mortality fluctuations caused by epidemics and other natural or man-made disasters. To address this interest and provide a basis for further research, in May 2020, the Short-term Mortality Fluctuations data series was launched as a new section of the Human Mortality Database. At present, this unique data resource provides weekly mortality death counts and rates by age and sex for 38 countries and regions. The main objective of this paper is to detail the web-based application for visualizing and analyzing the excess mortality based on the Short-term Mortality Fluctuation data series. The application yields a visual representation of the database that enhances the understanding of the underlying data. Besides, it enables the users to explore data on weekly mortality and excess mortality across years and countries. The contribution of this paper is twofold. First, to describe a visualization tool that aims to facilitate research on short-term mortality fluctuations. Second, to provide a comprehensive open-source software solution for demographic data to encourage data holders to promote their datasets in a visual framework.

## Introduction

The COVID-19 pandemic indicated the lack of publicly available data needed to trace the course of epidemics and provide a data-based ground for political decisions in response to short-term health challenges. The monitoring of the rapidly changing situation is a major challenge for statistical and public health systems during short term disasters. Data by causes of death can not be used for such monitoring because of differences in coding, approaches to testing in case of infectious diseases, and a long delay of data release. The causes of death statistics (mortality from influenza and other respiratory conditions) may also grossly understate the total death toll.

The excess mortality method is an important instrument to overcome these problems. The SARS-Cov-2 pandemic and other epidemics cause substantially more deaths than the officially registered deaths from COVID-19, influenza and/or all respiratory diseases. Many deaths

**Data Availability Statement:** The data are available from the Human Mortality Database (https://www.mortality.org). The source code repository is

accessible on GitHub (https://github.molgen.mpg.
de/nemeth/stmortality).

**Funding:** The work on the tool and the paper was supported by the Volkswagen Foundation (project "Strengthening a reliable evidence base for monitoring the COVID-19 and other disasters"). Support from the Basic Research Program of the National Research University Higher School of Economics is also gratefully acknowledged. The funders had no role in study design, data collection and analysis, decision to publish, or preparation of the manuscript.

**Competing interests:** The authors have declared that no competing interests exist.

happen due to the aggravation of pre-existing health conditions in the elderly. The method has been acknowledged by researchers and health authorities as the most reliable and objective way for the assessment of mortality elevation caused by the COVID-19 pandemic [1–4]. It is important to stress that this method is independent of the variability and the incomparability of diagnostics and causes of death coding across countries and time.

In May 2020, in response to the growing demand for such data, the Human Mortality Database (HMD, [5]) team launched the Short-term Mortality Fluctuations (STMF) data series. Currently, the STMF data series contains weekly death counts and death rates by age and sex for 38 countries and regions and is still growing. For an overview of the current coverage of the STMF see S1 Table. It is maintained jointly by the Max Planck Institute for Demo-graphic Research and the University of California at Berkeley under the aegis of the HMD, one of the most frequently used resources in demography.

The STMF provides data files in *xlsx* and *csv* formats, country- and region-specific metadata including a detailed description of data sources and data quality issues as well as a detailed description of the methodology applied. The STMF includes detailed data on death rates and counts that are given by broad age groups (0-14, 15-64, 65-74, 75-84, and 85+) and for all ages combined. Death rates are weekly rates, i.e. number of deaths per person-week.

We decided to add a visualization layer to the database to simplify the access to the data and enhance the basic understanding of the data for non-professional users. We also intended to facilitate and stimulate research on mortality outbreaks and seasonal variations for profes-sional users by providing a user-friendly tool for preliminary analysis. Finally, we would like to encourage dataset owners by providing our script creating this visualization tool in an online repository to showcase their data sources.

In the next section we provide a summary of the technical details and introduce the main features of the web-based tool.

## Materials and methods

The STMF visualization tool is an open-sourced, web-based `shiny` application for displaying excess mortality in weekly death counts or death rates. The tool can be accessed at https://mpidr.shinyapps.io/stmortality. The script is written in `R` programming language with the `shiny` [6], `data.table` [7], and `ggplot2` [8] packages with customizations added in `HTML`, `JavaScript` and `CSS` languages. We aimed to rely on the least amount of basic pack-ages for simplicity, compatibility and maintenance reasons. The repository is publicly available at the following web address: https://github.molgen.mpg.de/nemeth/stmortality.

The graphical user interface of the visualization tool can be divided into three main parts (see Fig 1). On the left-hand side, in the sidebar panel, different input variables can be selected to determine the method for calculation of excess mortality and to calibrate the figure shown under the title in the main graphical panel next to the sidebar. Summary statistical information on excess mortality can be acquired by interaction with this figure. These appear either in the figure overlaid or shown under the figure (e.g. in a table) after the user interacted with the fig-ure. The figures can be also stored as images after right-clicking on them.

The different functions of the web application are summarized in the User's Guide accessi-ble in the sidebar. In addition, the application has information buttons (represented by a blue letter i in bold) to give hints and explanations to the user of the various features on the website. These hints are revealed upon hovering with the cursor over the information buttons.

Links to the project's website on the HMD (under the title) and STMF directly to download data, metadata, and methodological notes are placed in the sidebar.

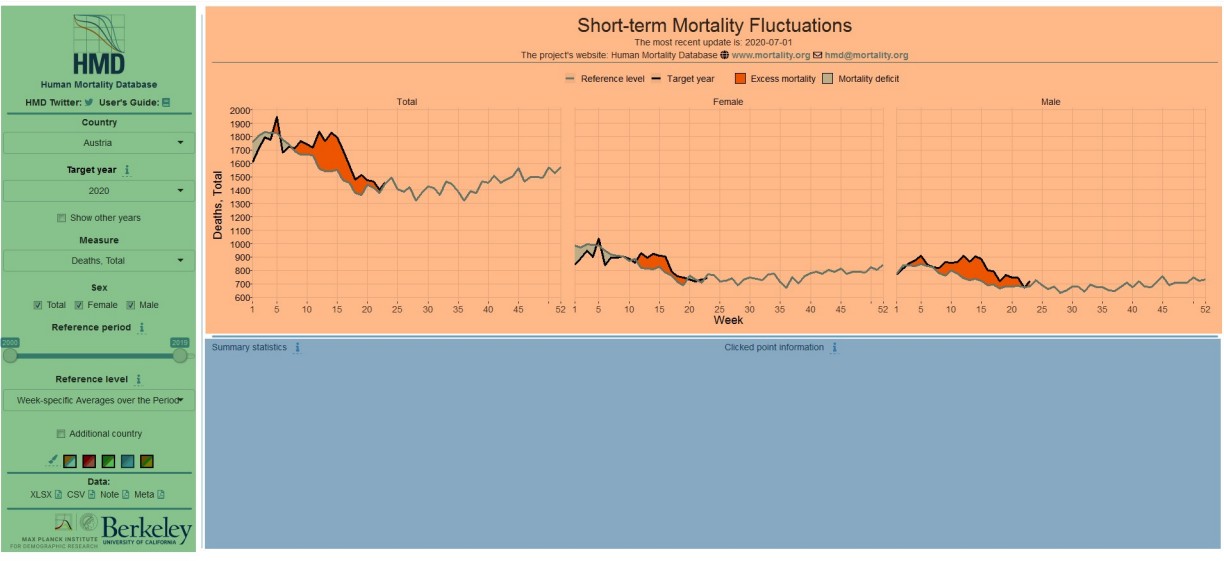

**Fig 1. The interface of the visualization tool.** The sidebar (indicated by green) contains the user input variables and links to the data source. The main figure appears in the area designated by orange color and summary statistical information is shown below the graphics in the blue area after user interaction.

## Input variables

A few input variables in the sidebar allow to determine how the main figure displaying excess mortality in the main panel is built. As a first step, a '*Country or region*' has to be selected out of the 38 countries and regions available in the STMF. The database provides death counts and death rates for all ages combined and by broad age groups, i.e., for 0-14, 15-64, 65-74, 75-84, 85+. One of these mortality measures has to be selected as the '*Measure*' variable. The user has to define the '*Sex*' variable as well, with its values being females, males and both sexes.

From the available years for the selected country or region a '*Target year*' can be selected. The target year is the main focus of interest and it is compared to a reference level to quantify the magnitude of the excess mortality in the target year, i.e., the difference between the target year and the reference level. The reference level is calculated for a user-defined reference period, a subset of the available years in the dataset. The next section, 'Estimation of excess mortality' provides details on the calculation of the implemented '*Reference level*' input variable.

It is possible to plot all available years for a country or region by clicking on the '*Show other years*' checkbox. When checked, the available years appear in light gray color and information on all data points are accessed by the click point information (see details in the 'Graphical and numerical presentation of excess mortality' section).

Selecting the '*Additional country or region*' checkbox reveals a new graphical panel below the main figure with the same features as the main figure and can be calibrated independently with input controls appearing additionally. This enables the user to compare two different countries and regions with each other or even different target years or reference levels, etc. of the same country or region.

With the help of the colored boxes next to the paintbrush icon the user can change the color scheme applied to the figures.

## Estimation of excess mortality

Direct and indirect estimations of excess mortality during wars and epidemics have a long history. In order to estimate the excess flu or heatwave deaths, researchers usually rely on the difference between the observed and expected (according to the usual conditions) number of deaths. The crucial methodological challenge is to estimate the expected mortality pertaining to a given week.

In the literature, there are two major approaches to the estimation of the intra-annual excess mortality [9, 10]. The first one is focused on the variation of mortality across weeks within a year in question and expresses a notion of "seasonality" [11–14]. The second one (actively used for assessment of the COVID-19-related mortality) investigates the mortality deviations for certain weeks compared to the mortality experience of previous years [10, 15–17]. This estimation depends on two components: a general mortality trend during the last few years and seasonal fluctuations. Such an estimation can be done using regression analysis [18] or the method of time series analysis [19, 20]. In the web application, based on various assumptions we implemented six different methods of the reference level for excess mortality estimation: the week-specific averages, the week-specific trends, the week-specific lower quartiles, the yearly average-week, the summer average-week, and the yearly lower-quartile-week. The first three methods determine reference levels corresponding to the second estimation approach, while the latter three correspond to the first approach. Nevertheless, this set should not be considered as exhaustive and we implemented them to present the variability of estimates.

Let $x_{ij}$ denote the value of the user-selected measure on week $i$ in year $j$.

The *Week-specific Averages* for week $i$, $\bar{x}_i$ equals the arithmetical mean over the years in the selected period and is given by

$$\bar{x}_i = \frac{\sum_{j \in P} x_{ij}}{|P|}, \tag{1}$$

where the set $P$ contains all years in the selected reference period and $|P|$ denotes the number of elements in the set $P$. The reference period may or may not include the target year.

For the target year $T$ and for week $i$, the reference level predicted by the *Week-specific Trends*, $\hat{x}_i$ is the expected value of a linear model estimated by ordinary least squares (OLS) over the years in the selected reference period. If there are 22 weeks available in the target year then the reference level consists of the results of 22 independent linear models. The estimation is carried out via the `lm` function of the `stats` package in R [21]. The reference level is given by

$$\hat{x}_i = \hat{a}_i + \hat{b}_i T, \tag{2}$$

where $\hat{a}_i$ and $\hat{b}_i$ are the estimated coefficients of the OLS regression over the reference period for week $i$.

The *Week-specific Lower Quartiles* for week $i$, $x_i^{Q_1}$ is the average of the values of the selected measure below the lower quartile in the reference period for week $i$. We implemented the calculation of the lower quartile with the help of the `quantile` function from the `stats` package. If there are 34 weeks available in the target year then this reference level requires the calculation of 34 lower quartiles, one for each week separately. Therefore, this reference level consists of values defined by

$$x_i^{Q_1} = \frac{\sum_{j \in L_i} x_{ij}}{|L_i|} \qquad L_i := \{j \in P : x_{ij} \leq quantile(\{x_{ij} : j \in P\}, 0.25)\} \tag{3}$$

The *Yearly Average-week*, $\bar{x}$ is the mean value of $\bar{x}_i$ values defined by Eq (1) for the available weeks in the target year and is calculated by

$$\bar{x} = \frac{\sum_{i \in W} \bar{x}_i}{|W|}, \tag{4}$$

where $W$ is the set of available weeks in the target year. Please note, if not all weeks are available in the target year, then $|W|$ could be lower than 52. As a consequence, $\bar{x}$ denotes the expected level of mortality if every week had the same average level of mortality in the year. This reference level, in contrast to the week-specific averages, reflects the seasonal variation of mortality within a calendar year or a number of years.

The *Summer Average-week*, $\bar{x}^*$ is similar to the above-mentioned yearly-average-week value, i.e. $\bar{x}$, but excludes from the calculation the winter weeks that tend to have higher mortality in general in comparison to summer weeks. For countries and regions in the Northern Hemisphere we define winter season as weeks from calendar week 1 to week 12 and from 48 to week 52 included, and weeks 22 and 38 for those situated in the Southern Hemisphere. Thus, generally, the value of $\bar{x}^*$ is expected to be lower than $\bar{x}$. The formula for the Summer Average-week is given by

$$\bar{x}^* = \frac{\sum_{i \in W^*} \bar{x}_i}{|W^*|} \tag{5}$$

with $W^*$ denoting the set of non-winter weeks, i.e., available weeks between calendar weeks 13 and 47 for the Northern Hemisphere and weeks 1 to 21 and 39 to 52 in the Southern Hemisphere.

The *Yearly Lower-quartile-week*, $\bar{x}^{Q_1}$ is the average of the values not greater than the lower quartile based on all the values of the user-selected measure in the reference period. This reference level requires the calculation of only one lower quartile value over a single year, e.g., for the target year, or over multiple years (lower quartile of the reference period). The Yearly Lower-quartile-week is defined by

$$\bar{x}^{Q_1} = \frac{\sum_L x_{ij}}{|L|} \qquad L := \{x_{ij} : x_{ij} \leq quantile(\{x_{ij} : j \in P\}, 0.25)\} \tag{6}$$

Eqs 1, 2 and 3 correspond to the mainstream approach that compares the observed week-specific mortality with that during a certain reference period. Correspondingly, the reference level of mortality varies across weeks. Eq 1 determines the reference level as the average over the reference period (following the excess mortality monitoring procedures by the Office of National Statistics in England and Wales, the New York Times, the Financial Times, the Our World in Data, etc.). In some countries (especially countries of Eastern Europe) mortality was steeply decreasing over the last 15-18 years. To address these trends, Eq 2 determines the baseline as a continuation of the week-specific trends. To address the heat-wave outbreaks and other mortality elevations out of the winter season, Eq 3 defines the reference level of mortality as the average mortality for the lower quartile of the mortality distribution across weeks within the reference period.

According to the mainstream practice by the Office of National Statistics in England and Wales and others, the reference period includes several years (usually 4-7 years) preceding the target year. For example, the years 2015-19 with 2020 as a target year. However, depending on the purpose, the reference period can be longer or shorter and may also include the target year (e.g. interpolation instead of extrapolation).

Eqs 4, 5 and 6 correspond to the "seasonality" approach. The excess mortality expresses losses due to the mortality difference between different periods within a year (or a number of years). Here, the reference level of mortality is a week-independent constant. In Eq 4, the reference level is defined as the average across all available weeks. In Eq 5, this level is defined as an average of mortality over a lower mortality season (non-winter season). In Eq 6, the low mortality season is defined in a flexible way according to mortality across weeks based on the lower quartile.

Eqs 4 to 6 may be used with the reference period consisting of the target year only. In this case, the excess mortality expresses the mortality elevation compared to the average mortality for all 52 weeks or to the average mortality over the lower-mortality weeks of the target year. However, if weekly mortality experiences large random fluctuations (in countries with small populations), the reference period could include the target year and a few years before this year, e.g., the years 2014-18 with 2018 as a target year.

## Graphical and numerical presentation of excess mortality

An important feature of the visualization tool is the possibility to estimate the intra-annual excess mortality. Upon opening the application, a country or region is presented with a focus on the excess mortality in 2020 compared to the reference level of week-specific averages over the whole available period of the previous years. There are three possible ways of interaction with the figure.

Firstly, hovering with the cursor over an interval that contains excess mortality (or deficit) polygon, information on this polygon is provided in a pop-up bubble next to the cursor, detailing the start and end weeks; and the length of this polygon. The example in Fig 2 gives information on a 25-week-long excess mortality polygon between weeks 1 and 25.

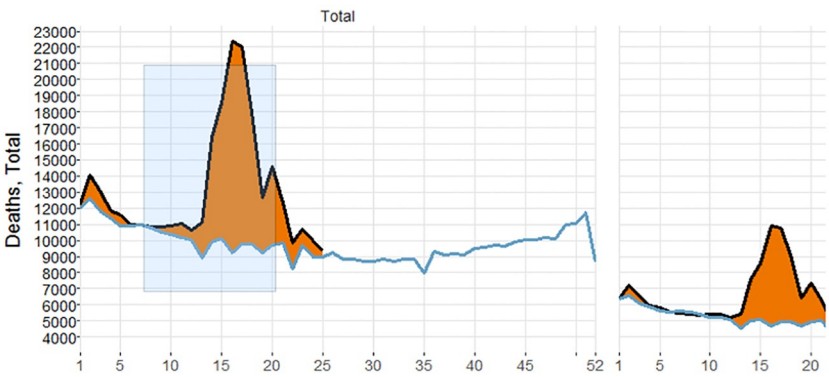

Summary statistics ℹ
The selection in England and Wales is between weeks 8 and 20 for the year 2020 compared to the period of 2010-2019.

| Sex | Difference | Deaths | Deaths, Target | Deaths, Reference |
|------|-----------------|--------|----------------|-------------------|
| Total | Excess mortality | 61275 | 189786 | 128511 |
| Female | Mortality deficit | -137 | 10781 | 10918 |
| Female | Excess mortality | 27855 | 82250 | 54394 |
| Male | Excess mortality | 33557 | 96755 | 63198 |

**Fig 2. Hover information of a polygon.** Information on the duration—the start, end and length—of an excess mortality polygon appears on hovering over a polygon.

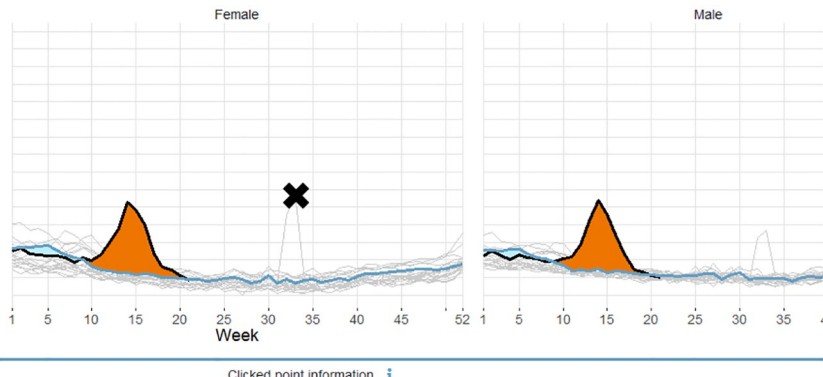

**Fig 3. Summary statistics information of a selection.** Information on death counts or death rates of the selected weeks by sex and type of difference are displayed below the figure.

Second, selection in the figure by brushing with the cursor reveals summary information below the figure. The sum of death counts or the mean of the death rates over the selected weeks are shown by sex and the type of the difference between the target year and the reference level. The example in Fig 3 shows a selection between weeks 8 and 20 for the year 2020 compared to the reference period of 2010-2019 for England and Wales. During these weeks England and Wales faced a total of 61275 excess deaths. Please note that the selection is week-dependent and summary statistics might not correspond to the exact area of the selection on the other axis, e.g., in this figure the tip of the excess mortality polygon is not selected.

Third, when the checkbox '*Show other years*' is selected clicking in the figure in the vicinity of a data point reveals information on the nearest data point. Fig 4 presents a click event for French females with target year 2020. The click point information identifies the data point

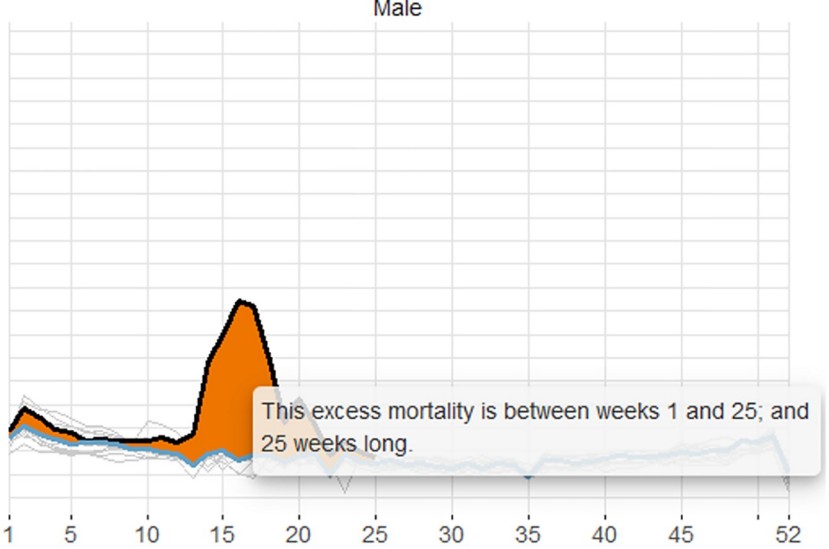

**Fig 4. Click point information.** Clicking in the figure helps identifying the nearest data point by revealing which year and week it pertains to.

from 2003 on week 33 that pertains to the unusually high amount of the total deaths due to the heatwave in that year. As a consequence, click point information can speed up outlier identification significantly.

All the above-mentioned interaction methods are available for the additional country or region panel, therefore, summary and click point information can be directly compared below the figures.

## Discussion and conclusion

The STMF is a new resource for filling the data gap in excess mortality estimation by providing both death rates and deaths. The basic features of the web-based visualization tool are already powerful enough to perform exploratory studies on the STMF data series and the simple layout of the visualization is transparent for use. Due to the open data policy of the STMF, our web application directly provides numerical values of death rates and excess mortality information for chosen periods within the target year. In addition, the web application enables the user to compare two country or region graphs with a flexible choice of the reference period and the measure of excess mortality. The implemented measures of intra-annual excess mortality provide a range of possible estimates and should help the users to form their own evidence-based opinion.

The tool has several limitations that are mainly direct consequences of the limitations of the STMF dataset. First, only countries or regions with high-quality population estimates and death statistics are included in the STMF. Therefore, the functionality of the web application is limited to these countries or regions, some of which provide a short time series only. Another limitation of our web application is the number of implemented definitions of excess mortality. We included only the most transparent and simple methods. We do not provide confidence limits for calculated quantities. Be that as it may, the customization and implementation of additional definitions of excess mortality is straightforward based on the source code available in the public repository.

The monitoring of the ongoing pandemic and the effectiveness of policy responses have been seriously affected by the lack of clearly defined criteria and reliable timely evidence based on the scale and course of the pandemic. The intra-annual excess mortality estimation provides an objective, reliable measurement that is clearly defined, easily understandable, and comparable across countries. This measurement has important advantages over COVID-19 morbidity and cause of death data comparisons suffering from multiple deficiencies related to differences in testing coverage and coding practices. In addition, the data on all-cause mortality is available after a reasonable short delay of few weeks while data on causes of death is published with a delay of one year or more. In this way, the visualization tool contributes to the dissemination of information to society and policy-makers about the real scale and course of epidemics by avoiding misleading information and speculations based on deficient data.

## Supporting information

**S1 Table. Supplementary table on the current status of the STMF dataset.** The table describes the time horizon the data is available for each country.
(PDF)

## Acknowledgments

The authors would like to thank Maayke de Boer, Magali Barbieri, Ainhoa Alustiza-Galarza, France Meslé and David Leon for the helpful comments on usage of the web-based tool. The

authors are grateful to Rainer Walke and Dirk Vieregg for facilitating the online framework for the web tool.

## Author Contributions

**Conceptualization:** László Németh, Dmitri A. Jdanov, Vladimir M. Shkolnikov.

**Methodology:** László Németh, Dmitri A. Jdanov, Vladimir M. Shkolnikov.

**Software:** László Németh.

**Visualization:** László Németh.

**Writing – original draft:** László Németh, Dmitri A. Jdanov, Vladimir M. Shkolnikov.

**Writing – review & editing:** László Németh, Dmitri A. Jdanov, Vladimir M. Shkolnikov.

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
