## [Decision Letter · Decision Letter 0]

22 Dec 2020

PONE-D-20-34864

An open-sourced, web-based application to analyze weekly mortality excess based on the Short-term Mortality Fluctuations data series

PLOS ONE

Dear Dr. Németh,

Thank you for submitting your manuscript to PLOS ONE. After careful consideration, we feel that it has merit but does not fully meet PLOS ONE’s publication criteria as it currently stands. Therefore, we invite you to submit a revised version of the manuscript that addresses the points raised during the review process.

This is a very important paper and contribution to the field and public health planning. The paper reads well and informs the reader about the database and how one is supposed to use it. As for an academic paper, based on the reviewers and my own reading, we believe it needs very minor adjustments. The main one is related to the calculation of excess mortality - please provide more detail. There are also some minor adjustments in language and several typos across the paper. I am looking forward to see the revised version. 

We look forward to receiving your revised manuscript.

Kind regards,

Bernardo Lanza Queiroz, Ph.D

Academic Editor

PLOS ONE

Journal Requirements:

Reviewers' comments:

Reviewer's Responses to Questions

**Comments to the Author**

1. Is the manuscript technically sound, and do the data support the conclusions?

Reviewer #1: Yes

Reviewer #2: Yes

Reviewer #3: Partly

2. Has the statistical analysis been performed appropriately and rigorously? 

Reviewer #1: N/A

Reviewer #2: Yes

Reviewer #3: I Don't Know

3. Have the authors made all data underlying the findings in their manuscript fully available?

Reviewer #1: Yes

Reviewer #2: Yes

Reviewer #3: Yes

4. Is the manuscript presented in an intelligible fashion and written in standard English?

Reviewer #1: Yes

Reviewer #2: Yes

Reviewer #3: Yes

5. Review Comments to the Author

Reviewer #1: This submission provides an overview of the new Short-term Mortality Fluctuations data series, available at the Human Mortality Database, and a new tool for visualizing excess mortality.

The paper is very informative and easy to understand, even for non-technical users. I have only a couple of minor comments, which should addressed in a revision:

• Main title, lines 113 and 126: You're juggling around with the terms "excess mortality" vs. "mortality excess". As the term "excess mortality" is commonly used in epidemiological studies, you should stick with it or, if applicable, "excess deaths".

• Line 7: please delete the URL from the brackets and stick to the common citation style of internet sources.

• Line 8: The "STMF" abbreviation should be put before "data series", not after, as that is not included in the abbreviation.

• Lines 10-11: Please delete the country listing. Instead, I would like to see a table in the appendix, which contains a line for each country with the information, for which time horizon the data is available in each country. This varies significantly between the countries and you could give for the users a quick overview, which would be useful for comparative international studies. This table would be a nice addition to the description you provide here.

• Line 13: Add an "s" to "resource".

• Line 43: Change "These appear" to "This appears"

• I would like to know in the discussion, why you don't also provide weekly data on cause-specific mortality and whether you plan to do so eventually. I know there's a report on the HMD homepage where you mention that as well. Please discuss this topic shortly in this contribution as well.

I suggest publication of the contribution after having addressed my points.

Reviewer #2: This is really great and I'm very thankful the teams at Max Planck and Berkeley have put this incredible resource together. As a long time user of the HMD, I just want to first say thank you!

I only have two minimal comments that should be very easy to address and do not preclude publication.

1) I'd like to see the authors include the calculations they are using for excess mortality in this manuscript.

2) There are a few minor typos throughout.

Again, thank you for providing such valuable resources!

Reviewer #3: An open-sourced, web-based application to analyze weekly mortality excess based on

the Short-term Mortality Fluctuations data series

A. Overview:

This article presents a novel tool for visualizing weekly excess mortality in 36 countries. Excess mortality provides us with valuable information during an epidemic or other natural or man-made disasters. In the context of the global COVID-19 pandemic, excess mortality estimations have been invaluable to evaluate the impacts of the pandemic and to compare territories. The tool presented is simple, yet it could be of vital importance to examine and compare the evolution of mortality in this and in other health emergencies and disasters worldwide. I really appreciated the simplicity and speed of the tool (other shiny apps can be quite slow). I think the article is interesting for a wide range of readers (researchers, demographers, public health professionals, policymakers etc.) and is suitable for publication in this journal; but I have some observations:

B. Major Comments:

1. The introduction is too short, and it fails to provide the necessary justification for the tool presented. A brief description of excess mortality and its usefulness for public health and policy will really help to show non-specialist readers the true importance of the data and visualizations provided. A proper contextualization of the COVID-19 pandemic and how excess mortality is helpful in this context can also be important.

2. As excess mortality is the main measure presented in the paper, further description of the different estimation methods (and why you choose the ones presented in the paper) is needed. For example, why did you decided to choose the average of the reference period and not the maximum historical value? What is the recommended reference period? Why eliminate winter months but keep summer months of years with clear outliers as European countries in 2003? A brief explanation on the methods will really help the reader to understand these questions.

3. When explaining the reference levels, you define a summer and winter seasons based on calendar weeks. This definition does not fit for countries in the southern hemisphere. Please correct or explain.

4. Both of the presented estimations (numerical excess deaths and crude death rates) do not allow comparisons between countries (because of population size, mortality trends and socio-demographical characteristics as ageing). This has to be addressed in the paper. Is there a way to compare territories? Can the excess mortality in percentage be useful?

5. The discussion is short and I fond that much can be said about the usefulness, limitations and strengths of the tool. Also, the reference to the fertility tool is not well connected to the rest of the article (how the fertility tool relates to short term mortality fluctuations?).

C. Minor Comments:

1. Why did you decided to include the target year in the linear models to estimate reference levels? Please explain.

2. When presenting death rates, it could be better to include the value in death per 10.000 or 100.000 persons/week. The interpretation of decimals can be hard for many readers.

3. The figures presented are low quality and do not reflect the beauty of the web tool.

4. The tool allows to change the colors of the graphs, which is helpful; but when displaying two graphs, it does not allow to change colors of each individual graph. For comparisons, this could be a neat feature.

6. PLOS authors have the option to publish the peer review history of their article (what does this mean?). If published, this will include your full peer review and any attached files.

Reviewer #1: **Yes: **Patrizio Vanella

Reviewer #2: No

Reviewer #3: **Yes: **Andrés Peralta Chiriboga

---

## [Author Response · Author response to Decision Letter 0]

22 Jan 2021

Dear Editor, Dear Reviewers, please see our responses in the Response to Reviewers file uploaded with the revision as requested by the editor in the decision letter.

---

## [Editor Report · Decision Letter 1]

25 Jan 2021

An open-sourced, web-based application to analyze weekly excess mortality based on the Short-term Mortality Fluctuations data series

PONE-D-20-34864R1

Dear Dr. Németh,

We’re pleased to inform you that your manuscript has been judged scientifically suitable for publication and will be formally accepted for publication once it meets all outstanding technical requirements.

Kind regards,

Bernardo Lanza Queiroz, Ph.D

Academic Editor

PLOS ONE

Additional Editor Comments (optional):

Regarding the inclusion of the statement about the funding, I believe the production team is responsible for adding the information. As the Academic Editor, I appreciate the information sent by the authors. I would like to congratulate the authors for this important contribution - not only the paper, but the public available database that is extremely relevant in the current situation. 
---

## [Editor Report · Acceptance letter]

27 Jan 2021

PONE-D-20-34864R1 

An open-sourced, web-based application to analyze weekly excess mortality based on the Short-term Mortality Fluctuations data series 

Dear Dr. Németh:

I'm pleased to inform you that your manuscript has been deemed suitable for publication in PLOS ONE. Congratulations! Your manuscript is now with our production department. 

Kind regards, 

on behalf of

Dr. Bernardo Lanza Queiroz 

Academic Editor

PLOS ONE